

# Dust deposition fluxes at the gateway to the Southern Ocean: investigating the use of lithogenic tracer measurements in aerosols collected in Tasmania, Australia

Claudia Hird [a*], Morgane M.G. Perron [ab*], Thomas M. Holmes [c], Scott Meyerink [c], Christopher Nielsen [a],

Ashley T. Townsend [d], Patrice de Caritat [e], Michal Strzelec [a], Andrew R. Bowie [ac]

a Institute for Marine and Antarctic Studies (IMAS), University of Tasmania, Battery Point, Tasmania, Australia.

b Université de Brest - UMR 6539 CNRS/UBO/IRD/Ifremer, Laboratoire des sciences de l'environnement marin (LEMAR) - Institut Universitaire Européen de la Mer - Rue Dumont D'Urville, 29280 Plouzané, France

c Australian Antarctic Program Partnership (AAPP), University of Tasmania, Battery Point, Tasmania, Australia.

d Central Science Laboratory, University of Tasmania, Hobart, Tasmania, Australia

e John de Laeter Centre, Curtin University, Bentley WA 6845, Australia

* These authors contributed equally to this work.

Correspondence to: Morgane M.G. Perron, morgane.perron@univ-brest.fr

## Abstract

Australia contributes a significant amount of dust-borne nutrients (including iron) to the Southern
Ocean, which can stimulate marine primary productivity. A quantitative assessment of the
variability of dust fluxes from Australia to the surrounding ocean is therefore important for
investigating the impact of atmospheric deposition on the Southern Ocean's carbon cycle. In this
study, lithogenic trace metals (aluminium, iron, thorium and titanium) contained in aerosols
collected between 2016 and 2021 from kunanyi/Mount Wellington in lutruwita/Tasmania
(Australia) were used to estimate dust deposition fluxes. Lithogenic fluxes were calculated using
each tracer individually, as well as an average using all four tracers. This latter approach enabled
an assessment of the uncertainty associated with flux calculations using only individual tracers.
Elemental ratios confirmed the lithogenic nature of each tracer in aerosols when compared with
both Australian soil samples and the average Earth's upper continental crust. Determined
lithogenic flux estimates were consistent with a regular dust deposition peak during the austral
summer, in line with the dust storm season in the southeast of Australian, and a low atmospheric
deposition in winter. This study provides an insight into the seasonal and interannual variability of





dust deposition fluxes from the southeast of Australia based on aerosol sample measurements. This
information will enhance our understanding of nutrient-bearing dust deposition to the Australian
sector of the Southern Ocean and may prove useful in refining modelling estimates of southern
hemisphere atmospheric deposition fluxes and their subsequent impact on global biogeochemical
cycles.

**21   Environmental significance / Plain language summary**

Dust deposition flux was investigated in lutruwita/Tasmania, Australia, between 2016 and 2021.
Results show that the use of direct measurement of aluminium, iron, thorium and titanium in
aerosols to estimate average dust deposition fluxes limits biases associated with using single
elements. Observations of dust deposition fluxes in the Southern Hemisphere are critical to
validate model outputs and better understand the seasonal and interannual impacts of dust
deposition on biogeochemical cycles.

## 28   1. Introduction

Lithogenic mineral particles such as iron oxyhydroxides, kaolinite, illite and smectite are
commonly entrained into the atmosphere (Cudahy et al., 2016) following the erosion of the Earth's
Upper Continental Crust (UCC) (Crawford et al., 2021). Such dust particles are the primary source
of trace metals including aluminium (Al), iron (Fe), thorium (Th) and titanium (Ti) to the
atmosphere, which can therefore be used as tracers of aeolian lithogenic inputs to the ocean (Baker
et al., 2020). Dust carries important nutrients, including Fe, to marine ecosystems, feeding primary
producers (Mackie et al., 2008). Due to the current lack of field observations on the concentrations
of aeolian trace metals and their corresponding dust deposition fluxes, large uncertainties remain
regarding how and to what extent dust supply fertilises key oceanic regions such as the Southern
Ocean with vital nutrients. This leads to a poor understanding of the impact of dust deposition on
the biological carbon pump.

The amount of dust entrained into the atmosphere depends on soil surface roughness, vegetation
and coverage, on particle size, composition, and moisture content, and on local conditions such as
wind speed and rainfall, which change both regionally and seasonally (Mahowald et al., 2009). Air
masses can carry dust over thousands of kilometres before particles return to land or fall onto the



surface ocean (Mackie et al., 2008). Atmospheric deposition of dust to the open ocean has been
demonstrated to act as a key supplier of vital macro- and micro-nutrients (such as Fe) to the marine
ecosystem (Mackie et al., 2008; Weis et al., 2024). For example, during the austral summer 2019-
2020, nutrient supply from large dust-containing bushfire emissions (Perron et al., 2022; Hamilton
et al., 2022) was identified as the main trigger of a large and long-lasting phytoplankton bloom in
the South Pacific Ocean (Weis et al., 2022).

Field and modelling approaches to estimating dust deposition both offer various benefits and
drawbacks. Field observations at sea are influenced by local environmental conditions (i.e.,
weather, surface ocean properties) and are not representative of the large scale or long-term
atmospheric deposition trends (Anderson et al., 2016). Time-series stations on land can overcome
the issue of temporal coverage but may not be representative of atmospheric loading and
deposition over remote oceanic regions. To date, global models are not capable of reproducing
atmospheric concentrations of trace metals transported in dust to remote areas and cannot
accurately quantify particle settling rates (Anderson et al., 2016). Considering the Southern
Hemisphere, model estimates tend to overestimate total dust emission at the source and
underestimate soluble trace element deposition fluxes over the ocean (Anderson et al., 2016; Ito et
al., 2020). To reduce uncertainty in dust deposition fluxes to the open ocean it is essential to
validate model fluxes using field-based observations. Long-term atmospheric observatories,
particularly near the coasts, are attracting increasing interest from the scientific community as a
platform to better understand seasonal to interannual patterns of deposition events in addition to
shipboard observations and satellite estimates (Perron et al., 2022; De Deckker, 2019).

In Australia, the large spatial heterogeneity of soil types and the highly episodic nature of weather
events such as droughts, bushfires and dust storms make it particularly difficult to model dust
deposition fluxes (Mackie et al., 2008). A main source of trace metals to the Australian sector of
the Southern Ocean is dust carried from kati thanda/Lake Eyre and dhungala-barka/Murray-
Darling geological basins (De Deckker, 2019). The typical dust storm season in Australia spans
from September to November (austral spring), with the most extreme storms occurring in
September (O'Loingsigh et al., 2017). The dust season can extend through the austral summer due
to bushfires (and postfire unvegetated ground) across southern Australia (Hamilton et al., 2022).



In a study conducted by Perron et al. (2022), atmospheric concentrations of mineral dust and
associated lithogenic tracers (Al, Fe and Ti) were reported to be 2.5-fold higher, on average, during
fire events compared to days not impacted by bushfires in lutruwita/Tasmania, Australia.

Dust deposition fluxes reported by different models range over an order of magnitude (from 0.55
to 5.48 mg m$^{-2}$ d$^{-1}$) over the Southern Ocean region southeast of Australia (Mahowald et al., 2006;
Weis et al., 2024). Different methods have been used to estimate dust deposition fluxes from the
analysis of a single tracer element, for example Al or Th, in aerosol samples and in seawater
(Anderson et al., 2016). However, single element dust flux estimates are subject to anomalous data
stemming from contamination, deviation from the mean UCC, or preferential mineralization
following a particular laboratory protocol. Recently, the analysis of four lithogenic tracers
(namely, Al, Fe, Th, and Ti) in marine sinking particles collected at the Southern Ocean Time-
Series (SOTS) mooring station (140°E, 47°S) were used to calculate an average 'multi-tracer'
estimate of dust deposition fluxes to surface waters of the subantarctic ocean south of Australia
(Traill et al., 2022). The latter field-based flux estimates showed good agreement with remotely
sensed proxies of dust transport and modelled deposition estimates. Elemental ratio analysis in the
same sediment trap samples suggested that lithogenic material from southeastern Australia was
the most likely source of Al, Fe, Th and Ti to this area of the Southern Ocean (Traill et al., 2022).

In this study, the analysis of the same four lithogenic tracers (Al, Fe, Th, and Ti) was performed
in aerosol samples collected at the kunanyi/Mount Wellington time-series sampling station in
southern lutruwita/Tasmania (Australia). Dust deposition fluxes were estimated from both
individual tracer concentrations and using the multi-tracer approach used by Traill et al. (2022).
Here we report a 5-year (2016-2021) time-series of dust deposition flux estimates downwind of
the south-eastern Australian dust path, at the gateway to the Southern Ocean. The suitability of the
four metals as lithogenic tracers was also verified by comparing elemental ratios (relative to Al)
in the aerosol samples to the average topsoil composition in Australia (this study) and to the
averaged UCC composition (McLennan et al., 2001).



## 2. Material and methods

### 2.1 Aerosol collection and study site

kunanyi/Mount Wellington overlooks Hobart, the capital city of the Australian island state of lutruwita/Tasmania. The mountain is in a strategic position for sampling one of the three major atmospheric pathways in Australia (Baddock et al. 2015; Bowler 1976), where air-masses from mainland Australia are transported south-eastwards over lutruwita/Tasmania (and our sampling site, Figure 1) before reaching the Southern Ocean. This study uses aerosol filters collected on a HiVOL 3000 air particulate sampler (Ecotech, Acoem, Melbourne, Australia) positioned at 1,271 m above sea level, on the summit of kunanyi/Mount Wellington. Filter samples have been collected for Total Suspended Particulates (TSP) since September 2016, with each sample representing a period ranging from a few days to 2 weeks, depending on weather conditions and specific weather events, and allowing for sampler servicing and calibration.

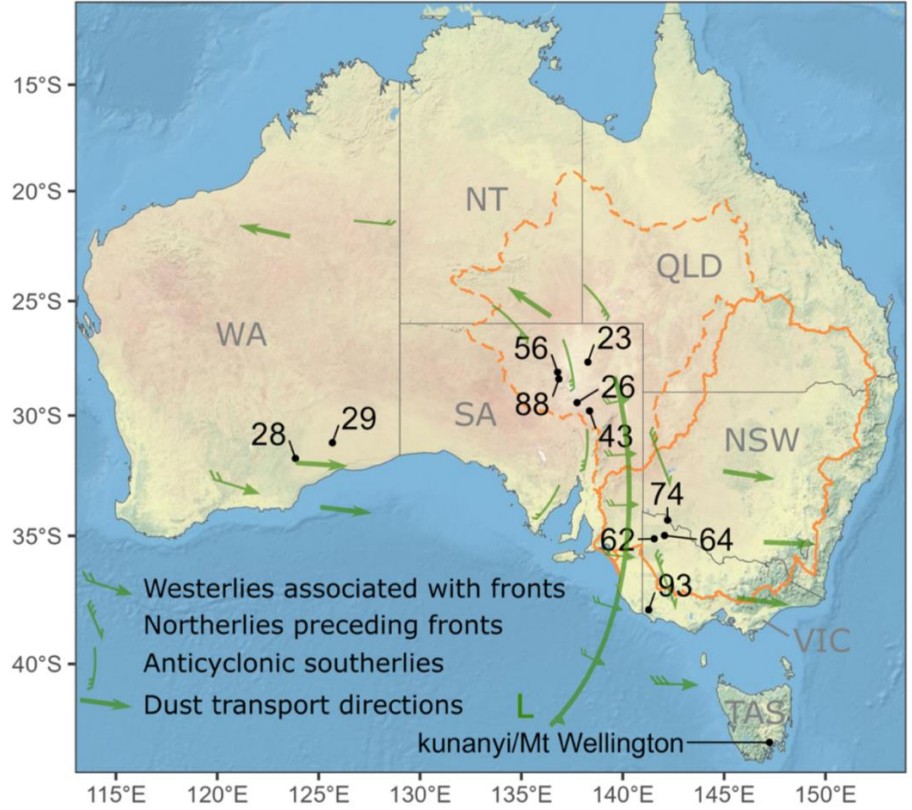





**Figure 1.** Location of the aerosol sampling station at kunanyi/Mount Wellington in Tasmania (TAS). Black
dots display the locations of selected NGSA soil samples in Western Australia (WA), South Australia (SA)
and Victoria (VIC) (see section 2.4) with their abbreviated identification numbers (see Table S2). Prevailing
wind pathways are also displayed as green arrows based on Spriggs (1982) and the kati thanda/Lake Eyre
(dashed line) and dhungala-barka/Murray-Darling (solid line) geological basins are delineated in orange.

2.2 Aerosol leaching protocol
Laboratory work for aerosol and soil sample processing (sections 2.2 and 2.3) followed
GEOTRACES recommended procedures for ultra-trace sampling and analysis (Cutter et al., 2017).
All reagents were ultra-high purity (UHP) and either purchased (Baseline, SeaStar Chemicals) or
distilled in-house using instrument quality reagents (IQ grade, SeaStar Chemicals). Whatman W41
(Sigma-Aldrich) filters were acid washed in a series of 24h hydrochloric acid (0.5 M HCl) baths
and rinsed with UHP water to leach any impurities and reduce the impact of the cellulose filter on
the analysis of trace elements in aerosols (Perron et al., 2020a).
Perron et al. (2020a) suggested a 3-step leaching method to define trace metal concentrations and
solubility in aerosols taken from land-based stations in Australia (Strzelec et al., 2020a, 2020b)
and on research vessels operating around Australia and in the Southern Ocean (Perron et al., 2020b,
2021). For this study, 125 aerosol samples were selected from the kunanyi/Mount Wellington
atmospheric particle time-series collection (November 2016 - February 2022). The origin and
concentration of aerosol Fe in 80 samples from this dataset was previously reported in Perron et
al. (2022), however the present study differs in using total concentrations of Fe, Al, Th and Ti to
calculate atmospheric (dust) deposition fluxes and the associated seasonal and interannual trends
at the sampling station. Although samples were collected and analysed in batch over several years,
the collection and analysis of each batch of samples follow the same protocol and the resulting
data was quality-controlled against blanks, replicate analysis and Certified Reference Materials
(Table S1).

Aerosol samples were successively leached using UHP water (Milli-Q®, 18.2 MΩ) and 1.1 M
ammonium acetate (10 mL, pH 4.7). The remaining filter residue was then digested using a mixture
of concentrated nitric acid ($HNO_3$, 1 mL) and concentrated hydrofluoric acid (HF, 0.25 mL) at
120°C for 12 hours (Perron et al., 2020a). The sum of all three steps in the protocol provided the
total concentration data for each lithogenic tracer in aerosols which is used in this study (Perron et
al., 2020a). Satisfactory recoveries (>80%) were obtained for Al, Fe and Ti when applying the



total metal digestion step of the protocol to two reference materials, the Arizona Test Dust (ATD)
(Morton et al., 2013) and the GeoPT13 certified Koeln loess (International Association of
Geoanalysts) (Potts et al., 2003) (supplementary Table S1). A smaller recovery of 73% obtained
for Th highlights the unique extraction and stability chemistry of the metal which our protocol is
not optimised for. Thorium concentrations are therefore likely to be underestimated in this study
as discussed in section 3.

2.3 Atmospheric deposition flux estimates
The total concentration of each lithogenic tracer in our samples was used to calculate single tracer-
dust deposition flux estimates. Due to the lack of necessary meteorological data to estimate particle
deposition velocities specific to our study site, a single coarse particle deposition velocity was
applied to trace metal-bearing dust deposition estimates based on the literature in similar study
regions (Baker et al., 2017; Perron et al., 2020b; Winton et al., 2015). In this study, "F(X)" denotes
the deposition flux estimate for the individual lithogenic tracer "X". F(X) (in mg m$^{-2}$ d$^{-1}$) was
obtained following equation (1):

167                                   $$F(X) = Cx * Vd \tag{1}$$

where X is the lithogenic tracer – Al, Fe, Th or Ti ; $C_x$ is the total metal concentration (ng m$^{-3}$) in
aerosols and $V_d$ is a constant deposition velocity of 2 cm s$^{-1}$. The single deposition velocity holds
uncertainty as it does not account for the specific particulate size in different aerosol samples or
for specific atmospheric conditions such as humidity and wind speed at the collection time (Baker
et al., 2016; Winton et al., 2016; Duce et al, 1991).

Single-tracer dust (lithogenic) deposition flux estimates, $F_{Lith(X)}$, were then calculated by dividing
F(X) by the average abundance ($[X]_{UCC}$, wt%) of the element X in the UCC as reported in
McLennan (2001); Al = 8.04%, Fe = 3.5%, Th = 1.07x10$^{-3}$%, Ti = 0.41% following equation (2).

177                                   $$F_{Lith(X) =} \frac{F(X)}{[X]_{UCC}} \tag{2}$$

While $F_{Lith(X)}$ estimates are solely based on the analysis of a single lithogenic tracer, a multi-tracer
dust deposition flux estimate, $F_{LithAv}$, was obtained by calculating the average of all four $F_{Lith(X)}$ for
each individual aerosol sample. Multi-tracer $F_{LithAv,}$ estimates were calculated using both the
reported average UCC composition (McLennan, 2001) and Australian soil measurements (see
section 2.4 in this study) as references for comparison.

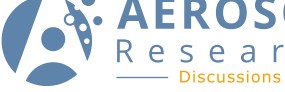


2.4 Soil sampling and processing
Eleven topsoil (0-10 cm) samples were selected from the National Geochemical Survey of
Australia (NGSA) Project: Geochemical Atlas of Australia (Geoscience Australia), a continental-
scale geochemical survey covering most of Australia (Caritat and Cooper, 2011; Caritat, 2022). In
this study, only selected soil samples originating from the Australian states of Western Australia,
South Australia and Victoria were analysed (Figure 1) as they likely better represent particles
entrained from the geological basins of kati thanda/Lake Eyre and dhungala-barka/Murray-
Darling, through the south-east Australian dust path towards our sampling station and the Southern
Ocean (Baddock et al., 2015, Supplementary Figure S1). It should be mentioned that no sample
from New South Wales was used for this study although a large part of the dhungala-
barka/Murray-Darling basin is located in this state.

A 10 mg aliquot of each soil sample was dry sieved through a 63 µm nylon screen to capture the
soil fraction fine enough to be entrained into the atmosphere (Strzelec et al., 2020a). The sieved
fraction was then processed through the same sequential leaching method described in section 2.2
(Perron et al., 2020a). Aerosol and soil leachates were analysed for a suite of elements, including
Al, Fe, Th and Ti, by Sector Field Inductively Coupled Plasma Mass Spectrometry (HR-ICP-MS,
Thermo Fisher Scientific, Element 2) at the Central Science Laboratory of the University of
Tasmania. Increased spectral resolution was employed to resolve major spectral interference
overlaps associated with analysis of Al, Fe and Ti. Further details on the ICP-MS analysis
procedure are provided in Perron et al. (2020a).

2.5 Atmospheric source tracking
The ratio between the total concentration of each lithogenic tracer of interest, T(X), and the total
Al concentration, T(Al), in individual aerosol samples was calculated and compared to the same
ratio in the average UCC reported in McLennan (2001) and in the average topsoil from
southeastern Australia (Section 2.3). The so-called enrichment factor (EF, equation 3) was used to
ascertain the lithogenic origin of Fe, Th and Ti in this study.
$$\mathrm{EF} = \frac{\frac{T(X)}{T(Al)}aerosol}{\frac{T(X)}{T(Al)}UCC} \qquad (3)$$

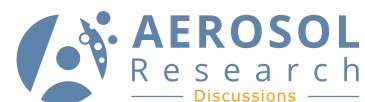

Using this approach, an EF value below 10 was considered to indicate a prevailing lithogenic
source origin for the metal tracers, while an EF exceeding the threshold value of 10 is associated
with an enrichment from non-lithogenic atmospheric sources such as anthropogenic combustion
(Shelley et al., 2015; Perron et al., 2022). Reimann and Caritat (2005) warned about the biases
associated with using a low EF threshold to identify anthropogenic sources due to the natural
variability in the Earth's crust composition, fractionation of elements during their emission to –
and transport within – the atmosphere, and biogeochemical processes during and after aeolian
transport. Here, a high EF threshold of 10 is adopted to account for such variability.

## 3. Results and discussion

### 3.1 Evaluating the lithogenic origin of the four tracers in aerosols

Enrichment Factors (EF) were calculated for Fe, Th and Ti measured in aerosols, and compared to
the Australian soil samples selected from the NGSA (this study) and compared to averaged UCC
composition from McLennan (2001) (Table 1). Calculated EF values were used to discard
significant contributions of non-lithogenic sources to our lithogenic tracers in kunanyi/Mt
Wellington aerosols as indicated by EF>10. Metal concentrations in individual NGSA soil samples
analysed in this study are reported in the supplementary Table S2.

**Table 1.** Comparison of mean Al, Fe, Th and Ti concentrations measured (ng mg$^{-1}$) in Australian soil samples (n = 11) compared to concentrations reported in the average UCC by McLennan (2001). Enrichment factors (EFs) calculated for Fe, Th and Ti (using Al as a reference) in aerosols collected at kunanyi/Mount Wellington (n = 125) are also displayed using both crustal references

|     | UCC   | Australian soils |        | kunanyi/Mount Wellington aerosols | |
| --- | ----- | ---------------- | ------ | ------ | ----------------- |
|     |       |                  |        | /UCC   | /Australian soil  |
| **Al** | 80400 | 38560 |        |        |                   |
| **Fe** | 35000 | 22616 | **EF(Fe)** | 1.6 ± 0.6 | 1.2 ± 0.5 |
| **Th** | 10.7  | 10.3  | **EF(Th)** | 1.3 ± 2.5 | 0.7 ± 1.2 |
| **Ti** | 4100  | 4313  | **EF(Ti)** | 1.2 ± 0.6 | 0.6 ± 0.6 |


Overall, EFs close to 1 were measured for all aerosol samples, suggesting that the lithogenic tracers
used in this study are indeed of a prevailing crustal origin. Using Australian soil concentration
(Table 1 and supplementary Table S2) to calculate EFs resulted in values further away from the



threshold of 10. In particular, EFs calculated using Australian soil data are closer to 1 for Fe and
Th when compared to using average UCC values (McLennan, 2001). Indeed, underestimated Th
measurements due to incomplete sample digestion (section 2.2) in our study result in a similar
underestimate of EF. Elemental ratio of Ti/Al in aerosol samples collected at kunanyi/Mt
Wellington (Figure 2) were closer to the average ratio of the UCC, resulting in EF(Ti) closer to 1
when compared to using average Australian soil measurement as a reference.

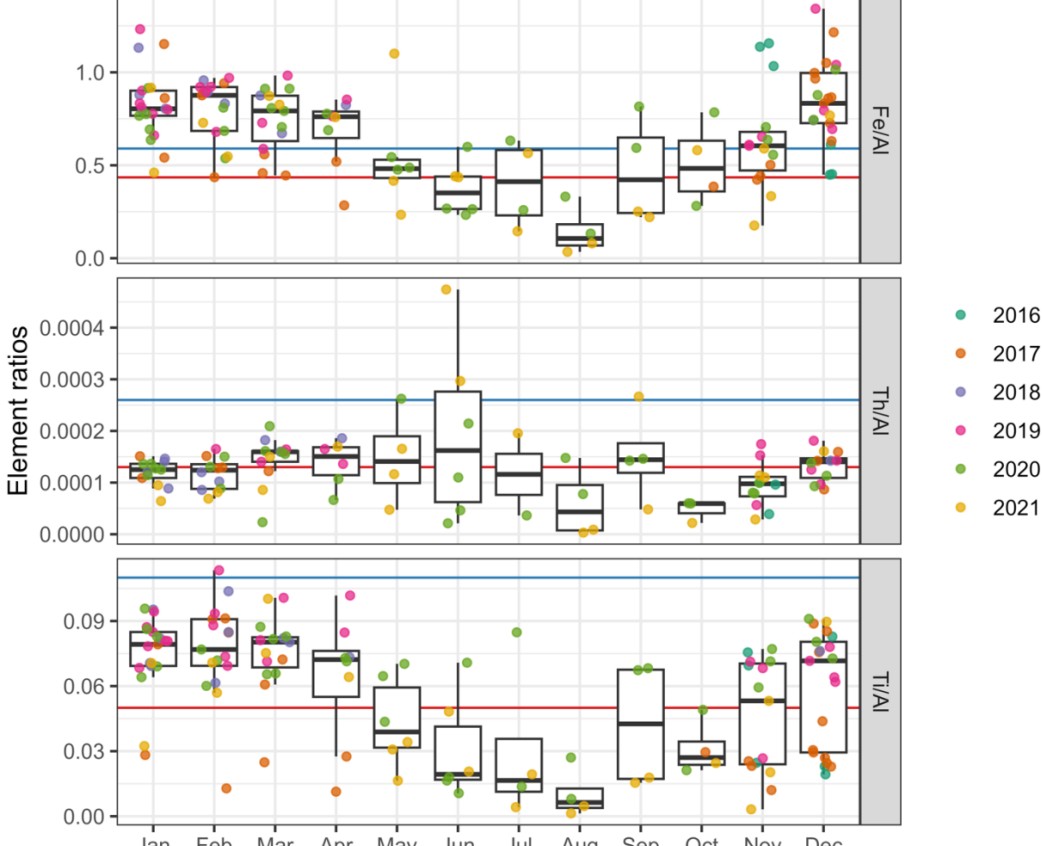

**Figure 2.** Boxplot of elemental ratios of Fe/Al (top), Th/Al (middle), and Ti/Al (bottom) in kunanyi/ Mt
Wellington aerosol samples collected between 2016 – 2021, grouped according to month. Whiskers
represent 1.5 times the interquartile range (75th – 25th percentile) beyond the boxes, while the upper, middle,
and lower horizontal lines of the box represent the higher interquartile, median value and lower interquartile
of the average monthly dataset, respectively. Colours represent each sample collection year. Horizontal red
lines represent metal ratios in the average UCC (McLennan, 2001). Horizontal blue lines represent average
metal ratios in the eleven selected NGSA Australian soil samples (this study). Two Th outliers (May and
June 2020) were excluded from the Th dataset and subsequent calculation for clarity.




Mean Al and Fe concentrations measured in our Australian soil samples were both twice smaller
than the average UCC values reported by McLennan (2001) while Ti and Th concentrations were
similar within 10% (Figure 2 and supplementary Table S2). While Australian soil is known for its
high Fe content (Mahowald et al., 2019, Strzelec et al., 2020a), a high soil heterogeneity across
this vast country may explain such surprising observation. This resulted in calculated Th/Al and
Ti/Al ratios significantly higher for Australian soil samples while Fe/Al ratios remained similar
compared to the average UCC.

Elemental ratios calculated for individual aerosol samples are summarised in the supplementary
Table S3. Both Fe/Al and Ti/Al ratios showed a clear seasonal trend, with higher ratios resembling
mean ratios measured in Australian soil samples (Fe/Al=0.59 and Ti/Al=0.11, Figure 2) in the
summertime (December – March) and lower Fe/Al and Ti/Al ratios closer to the average UCC
ratios (Fe/Al=0.435 and Ti/Al=0.05, McLennan, 2001) in wintertime (June – September, Figure
2). Summertime Fe/Al ratios in kunanyi/Mt Wellington aerosols were slightly higher (Fe/Al =0.72)
than the Australian soil measurements. This can be explained by increased contribution of local
soil emission from Tasmania under drier weather conditions and from postfire barren ground.
Indeed, the NGSA database shows high Fe/Al ratio (on average 0.7, n=21 samples) in soil samples
collected in Tasmania and processed through aqua-regia mineralization and x-ray florescence
analysis (Caritat and Cooper, 2011; Caritat, 2022). Ti/Al ratios were found to lie between our
Australian soil (Ti/Al = 0.11) and UCC (Ti/Al = 0.05) references from December through to May,
then falling below the UCC ratio in the cooler months of the year. This monthly variability
indicates different lithogenic sources of Fe and Ti are likely to influence the atmospheric loading
at our sampling station throughout the year. The onset of the dust season on the Australian
mainland (October-November, Baddock et al., 2015) may explain part of the summer (dusty)
season atmospheric inputs at our kunanyi /Mt Wellington aerosol sampling station, as evidenced
by higher Fe/Al (and Ti/Al) ratios in aerosols. On the other hand, other atmospheric sources
(locally derived from Tasmania or from long-range transport over the Southern Ocean) with a
similar (lower) metal/Al signature than the UCC seem to prevail in our study region during winter.
However, the small number of aerosol samples available between May - October in our study does
not allow for accurate assessment of trends during the winter period. Much smaller variability was



observed for the Th/Al ratio calculated in kunanyi/Mt Wellington aerosols (mean Th/Al = 0.00017)
across the time-series, with an overall median ratio close to that of the UCC (mean Th/Al =
0.00013) across most of the year except during August and October.

Differences between elemental ratios in soil and in aerosol samples may stem from atmospheric
processes occurring during transport between source regions and the sampling site including the
preferential settling of denser (e.g., oxyhydroxides) minerals over lighter minerals (e.g., clay), and
from the mixing of different lithogenic air-masses during atmospheric transport. Analysis of a
large set of soil samples, including more locations across Australia and particularly in Tasmania,
as well as high resolution information on wind speed and direction at the sampling site and for the
duration of the timeseries is necessary to better assess the relative contribution of different
Australian dust sources to the lithogenic particulate loading at kunanyi/Mount Wellington.

3.2 Single tracer lithogenic particle fluxes at kunanyi/Mount Wellington: characteristics
and trends
Thorium and Ti are commonly used as tracers of lithogenic atmospheric deposition fluxes as they
are almost exclusively derived from lithogenic material and have little reactivity or biological
utility in the atmosphere (Boës et al., 2001; Ohnemus and Lam, 2015). While Al may be emitted
to the atmosphere by anthropogenic sources, its prevailing source in the offshore atmosphere
remains crustal material (Xu and Weber, 2021). Although Fe solubility vary following physico-
chemical processes during the atmospheric transport, the soluble Fe fraction remains small
compared to the total (mostly refractory) fraction of Fe delivered by dust. Hence, if all four tracers
have a unique lithogenic source, the use of a multiple tracer lithogenic flux estimate can reduce
the uncertainty associated with the variability of a single metal's concentration due to
contamination, deviation from the UCC or secondary atmospheric inputs (Traill et al., 2022).

**Table 2.** Correlation coefficient ($R^2$) between tracer concentrations in kunanyi/Mount Wellington aerosols.

|        | Al   | Th   | Fe | Ti |
|--------|------|------|----|----|
| **Al** | 1    | -    | -  | -  |
| **Th** | 0.90 | 1    | -  | -  |
| **Fe** | 0.87 | 0.82 | 1  | -  |



| **Ti** | 0.74 | 0.84 | 0.83 | 1 |
|---|---|---|---|---|


A strong correlation (mostly $R^2 > 0.8$) was found between the total atmospheric concentrations of
Al, Fe, Th and Ti in the individual samples (Table 2). The strongest correlation ($R^2 = 0.90$) was
found between Al and Th and the weakest correlation ($R^2 = 0.74$) was found between total Ti and
Al concentrations in aerosols. Such strong correlations suggest that a common prevailing source
may supply all four tracers to kunanyi/Mt Wellington sampling station. Australian soil samples
collected in the state of Victoria and analyzed in this study also showed a good correlation between
the four lithogenic tracers, with $R^2$ of 0.97, 0.74 and 0.73 for Fe, Th and Ti when compared to Al
(based on Table S3 data). No significant correlation was ound for soil samples from Western
Australia and South Australia. However, the small number of soil sample analysed in this study
(n=4 for the state of Victoria) is not sufficient to draw conclusion on the potential origin of metals
in kunanyi/Mt Wellington aerosol samples. Indeed, the larger NGSA database available from
Caritat and Cooper (2011) only shows a meaningful correlation (>0.70) between Fe and Al
measurements in soil samples collected in New South Wales, South Australia and Tasmania using
an aqua regia mineralization and x-ray florescence analysis protocol.

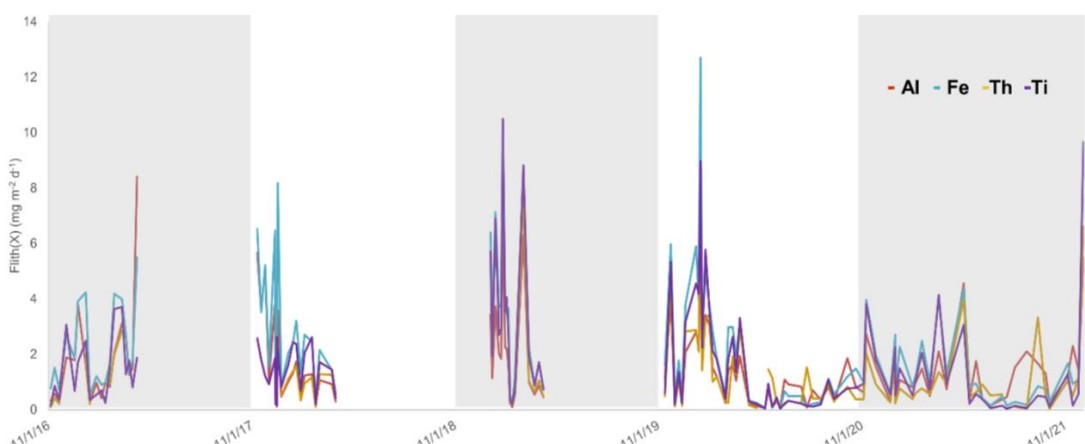


**Figure 3.** Individual tracer flux, $F_{Lith(X)}$ (mg m$^{-2}$ d$^{-1}$), at the kunanyi/Mount Wellington aerosol sampling
station from 2016 to 2021. Data points represent each aerosol mid-sampling period. Gaps in the time series
are periods where samples were not collected due to logistical limitations (winters) or instrument
maintenance. Shading denotes every other year starting on 1$^{st}$ of November each year. Here, $F_{Lith(X)}$ are
calculated using the average UCC content for each metal as reported in McLennan (2001).



Dust deposition fluxes estimated using individual tracer (Al, Fe, Th, and Ti) concentrations
measured in kunanyi/Mount Wellington aerosols, called $F_{lith(X)}$, showed similar variability
throughout the time-series (2016-2021, Figure 3). Overall, the smallest $F_{lith(Th)}$ flux was estimated
using Th as a single lithogenic tracer, ranging between 0.03 and 7.8 mg m$^{-2}$ d$^{-1}$. The largest dust
flux was obtained using Fe as a lithogenic tracer and ranged between 0.05 and 12.7 mg m$^{-2}$ d$^{-1}$
($F_{lith(Fe)}$, Figure 3). Lithogenic flux estimates calculated using Al and Ti concentrations in aerosols
ranged from 0.06 - 8.4 mg m$^{-2}$ d$^{-1}$ and from 0.03 - 10.5 mg m$^{-2}$ d$^{-1}$, for $F_{lith(Al)}$ and $F_{lith(Ti)}$,
respectively. Despite slight differences found between $F_{lith(X)}$ estimates obtained using different
lithogenic tracers, the magnitude of the difference between the highest and lowest $F_{lith(X)}$ estimates
varied by only a factor of 2, which reinforces the likelihood of a common prevailing atmospheric
source for all four tracers.

This finding corroborates work presented by Traill et al. (2022), where concentrations of all four
lithogenic tracers showed similar variabilities in marine sinking particles collected in the
subantarctic region of the Southern Ocean south of Tasmania (SOTS station). Similarly, Traill et
al. (2022) estimated higher lithogenic fluxes when using Fe as a lithogenic tracer and lower
lithogenic fluxes when using Th as a lithogenic tracer (Traill et al., 2022). Median $F_{lith(X)}$ estimates
measured at the kunanyi/Mt Wellington sampling site (this study: 1.2, 1.7, 0.8 and 1.1 mg m$^{-2}$ d$^{-1}$
using Al, Fe, Th and Ti as individual lithogenic tracer, respectively) compares well with reported
dust deposition fluxes of 1.4 - 5 mg m$^{-2}$ d$^{-1}$ estimated by models in the study region (Jickells et al.,
2005; Weis et al., 2024) and other Southern Hemisphere dust fluxes <2.7 mg m$^{-2}$ d$^{-1}$ reported off
the coasts of South Africa and South America, away from major dust sources (Menzel Barraqueta
et al., 2019). Our flux estimates are smaller than mineral dust deposition estimates of 4.0 - 25.0
mg m$^{-2}$ d$^{-1}$ (based on Ti concentration in aerosols) reported by Strzelec et al. (2020a) in Western
Australia, much closer to large Australian deserts.

Overall, maximum $F_{Lith(X)}$ estimates in our study were calculated during austral summer months
(roughly December – March). Different metals are observed to dominate the summer $F_{Lith(X)}$ peak
each year, with Al showing the highest $F_{Lith(X)}$ flux in summer 2016/17 (8.4 mg m$^{-2}$ d$^{-1}$), Fe in
2017/18 (8.2 mg m$^{-2}$ d$^{-1}$) and in 2019/20 (12.7 mg m$^{-2}$ d$^{-1}$), and Ti in 2018/19 (10.5 mg m$^{-2}$ d$^{-1}$)
and in 2021/22 (9.6 mg m$^{-2}$ d$^{-1}$). This may be due to variabilities in the nature and composition of



the dominant dust source arriving at the sampling site each year, including the impact of dust-
containing fire emissions during the summer seasons 2018/19 and 2019/20 (Perron et al. 2022).

3.3 Multi tracer particle flux
All four tracers (Al, Fe, Th, and Ti) measured in kunanyi/Mount Wellington aerosols showed a
strong correlation with one another and a similar variability over time (section 3.1), suggesting
that they originated from a single terrestrial source. This supports the approach taken in this study
whereby a multi-tracer lithogenic deposition flux, called $F_{LithAv}$, is estimated by averaging $F_{Lith(x)}$
fluxes obtained using each of the four tracers for each sample. The resulting $F_{LithAv}$ estimated at
our station between 2016 and 2021 is displayed in Figure 4 and provides a more robust estimate
of deposition flux by smoothing variability between tracers (displayed in Figure 3). Individual and
average lithogenic flux estimates ($F_{Lith(x)}$ and $F_{LithAv}$, respectively) calculated in this study are
summarised for individual samples in the supplementary Tables S4 and S5, respectively.

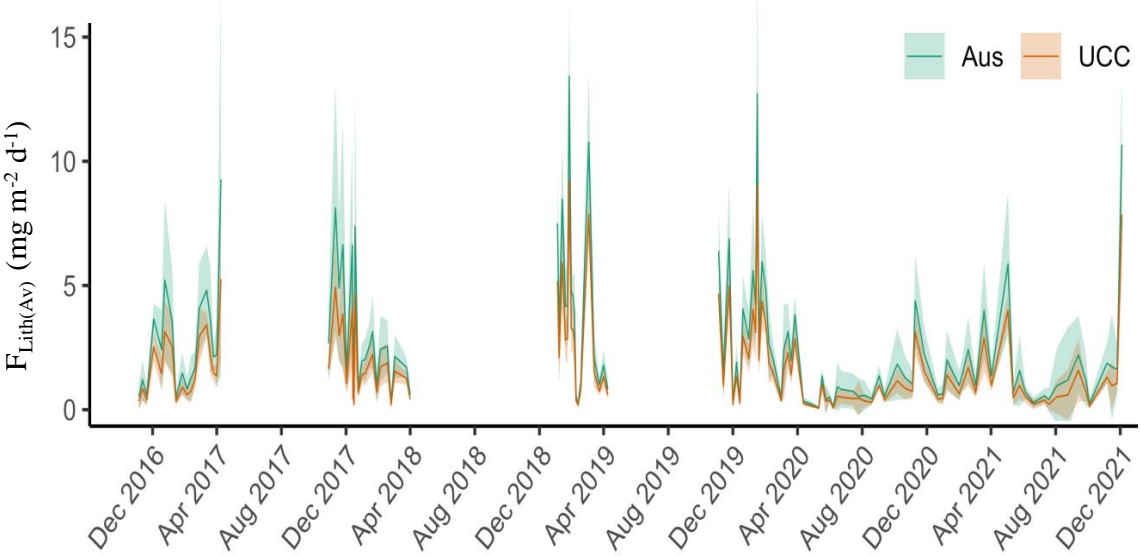


**Figure 4.** Multi-tracer lithogenic flux estimate, $F_{lithAv}$, expressed in mg m$^{-2}$ d$^{-1}$, corresponding to the average
of all individual tracer fluxes ($F_{lith(X)}$) calculated based on the lithogenic composition of the UCC (orange

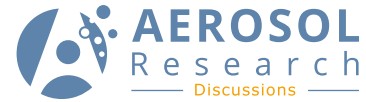



colour) and that of the eleven Australian soil samples measured in this study (green colour) . Shadings
represent +/- one $F_{LithAv}$ standard deviation of the average (solid lines).

A mean $F_{LithAv}$ value of $1.8 \pm 1.3$ mg m$^{-2}$ d$^{-1}$ was calculated based on the analysis of aerosol samples
collected between 2016 and 2021 at kunanyi/Mt Wellington (Tasmania, Figure 4 orange colour).
Throughout our time series, the highest $F_{LithAv}$ values were observed in January 2019 and 2020,
with flux peaks reaching 9.2 in January 2019 and 9.0 mg m$^{-2}$ d$^{-1}$ in January 2020, respectively.
Noticeable peak fluxes of 7.9 mg m$^{-2}$ d$^{-1}$ also occurred in early March 2019 and in mid-December
2021. Extended periods of low $F_{LithAv}$ estimates ($\leqq 0.5$ mg m$^{-2}$ d$^{-1}$) were observed during the two
austral winter periods sampled, with a minimum flux of 0.06 mg m$^{-2}$ d$^{-1}$ reached in May 2020
(Figure 4). There is therefore an apparent seasonal trend in dust deposited at the kunanyi/Mt
Wellington site, with higher $F_{LithAv}$ observed in warmer periods (November - March) and lower
fluxes in cooler periods of the year (May - August). It should be mentioned that a mean $F_{LithAv}$
value of $2.7 \pm 1.9$ mg m$^{-2}$ d$^{-1}$ is estimated when using the average metal content in Australian soil
analyzed in this study (Figure 4 green colour). Indeed, while Th and Ti contained in our eleven
Australian soil samples show similar concentrations (within 10%) as in the average UCC
(McLennan, 2001), Al and Fe concentrations in these local soil samples differ by 52 and 35%,
respectively. This result in higher $F_{LithAv}$ estimated using Australian soil data (Figure 4).
The mean $F_{LithAv}$ observed in this study, of 1.8 mg m$^{-2}$ d$^{-1}$ when using the average UCC and 2.7
mg m$^{-2}$ d$^{-1}$ when using the average Australian soil measurement (Table S3), fall within the dust
deposition range of 1.1 - 5.5 mg m$^{-2}$ d$^{-1}$ reported by models in southeastern Australia, which
account for soil erodibility, soil particle size distribution and wind friction velocity (Albani et al.,
2014; Weis et al., 2024). In the Southern Ocean south of Tasmania, smaller mineral dust fluxes of
0.37 mg m$^{-2}$ d$^{-1}$ and 1.0 mg m$^{-2}$ d$^{-1}$ were reported based on aerosol Fe measurements at sea, particle
size and surface wind speed (Bowie et al., 2009) and based on Al, Fe, Th and Ti measurements in
marine sinking particles (Traill et al., 2022), respectively. Traill et al. (2022) reported a similar
annual variability in lithogenic deposition flux at SOTS between 2011 and 2018, with minimum
$F_{LithAv}$ around 0.5 mg m$^{-2}$ d$^{-1}$ in July-September and an earlier dust deposition peak (compared to
our study) in November-December, up to 2.5 mg m$^{-2}$ d$^{-1}$. Strzelec et al. (2020a) also reported (up
to 6 times) higher mineral dust fluxes in warmer months compared to cooler months based on Ti
analysis in aerosols from Western Australia. In particular, the two summer seasons showing $F_{LithAv}$



over 9.0 mg m$^{-2}$ d$^{-1}$ correspond to large bushfire seasons in Tasmania and in Australian mainland
upwind from Tasmania, respectively (Perron et al., 2022). Indeed, fire events are known to
exacerbate dust entrainment into the atmosphere both during (pyro convective updrafts) and post
(burnt ground) fire event (Hamilton et al., 2022). It is worth noting that F$_{LithAv}$ estimated using
Australian soil measurements (this study) fall closer to the mean reported estimate found in the
literature while using the average UCC value result in lower-end F$_{LithAv}$ estimate compared to the
literature. While F$_{LithAv}$ estimated using the average UCC may present an advantage in being more
comparable with other studies worldwide, F$_{LithAv}$ estimated using Australian soil data may be more
relevant for validating model outputs as it likely better represents true deposition fluxes in our
study region.

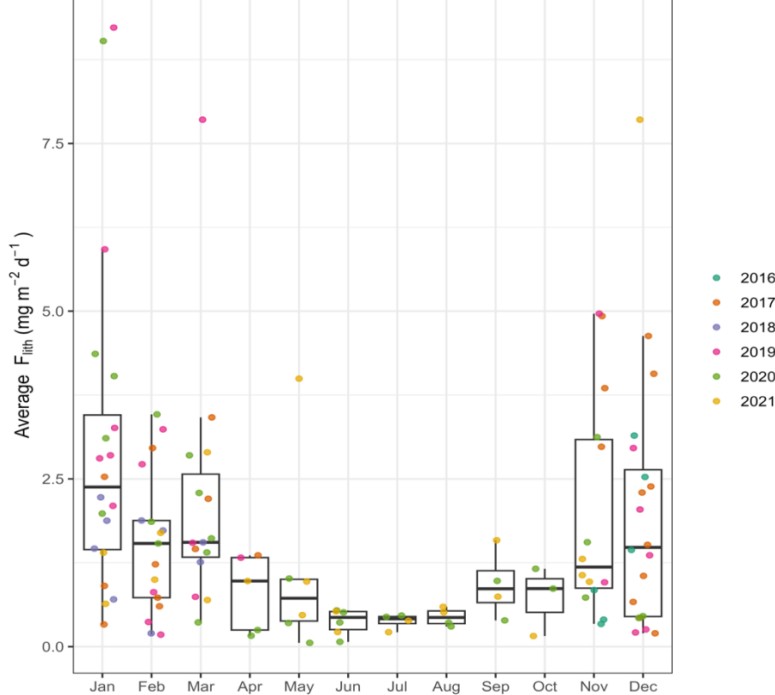

**Figure 5**. Monthly F$_{LithAv}$ estimates, in mg m$^{-2}$ d$^{-1}$, based on lithogenic tracer analysis in aerosol samples
collected between 2016-2021 at the kunanyi/Mt Wellington site. Individual (weekly) samples are shown as
dots and the colour code represents each collection year. Whiskers represent 1.5 times the interquartile
range (75$^{th}$ – 25$^{th}$ percentile) beyond the boxes, while the upper, middle, and lower horizontal lines of the



box represent the higher interquartile, median value and lower interquartile of the average monthly dataset,
respectively.
Greatest $F_{LithAv}$ fluxes are annually observed during the austral summer (December - March),
with median $F_{LithAv}$ of 2.4 mg m$^{-2}$ d$^{-1}$ in January and around 1.4 mg m$^{-2}$ d$^{-1}$ in December,
February and March across all years (Figure 5). This tendency aligns with higher frequency of
dust storms occurring in Australia's main geological basins during warmer months (late austral
spring and summer), resulting in higher dust deposition fluxes (O'Loingsigh et al., 2017). The
summers of 2017/2018 (Nov-Dec), 2018/19 (Jan-Feb) and 2019/20 (Dec-Feb) had especially
high $F_{LithAv}$ fluxes compared to other summer periods in the time-series (Figure 5). These
observations are consistent with the year 2017 being identified as the third driest year since
records have been kept in Australia (Steffen et al., 2018), and the two following summer periods
being identified as strong bushfire years, across Tasmania in 2018/2019, and across southeast
Australia in 2019/2020 (Perron et al., 2022). Relatively smaller peaks were observed during the
summer of 2020/21 and, to a lesser extent, during the 2016/17 summer (Figure 5). This may
reflect two wetter summer periods under the influence of El Niño Southern Oscillation positive
phase (La Niña), where increased moisture in the topsoil restricted particles from being eroded
and entrained by air masses (Bureau of Meteorology, 2022). In addition, fewer bushfire
emissions during these two wetter summer periods may have resulted in less dust emissions due
to increased vegetation cover on the soil (Bureau of Meteorology, 2022). Wetter summer seasons
may also explain a shift in $F_{lithAv}$ peaks towards the end of the summer seasons 2016/17 and
2020/21 (February - March) compared to the December-January $F_{lith(Av)}$ peak observed in
2017/18, 2018/19, and 2019/20 (Figure 5).

## 4. Conclusions
This study explores seasonal and interannual variability of the lithogenic deposition flux using
analysis of four lithogenic tracers (Al, Fe, Th, and Ti) in aerosol samples collected at kunanyi/Mt
Wellington (Tasmania, Australia). First, enrichment factors close to 1 and elemental ratios similar
to those measured in soil samples collected in Australia dust source regions enabled to confirm the
crustal origin of all four tracers. Deposition fluxes, $F_{Llith(X)}$, which were then calculated using each
tracer individually (X : Al, Fe, Th, or Ti) showed highly similar variability between one another





throughout the 2016-2021 time series. The small difference, of a factor 2 on average, observed
between the highest $F_{Llith(X)}$ (Fe as a lithogenic tracer) and the lowest $F_{Llith(X)}$ (Th as a lithogenic
tracer) estimates supported the development of an averaged lithogenic deposition flux, $F_{lithAv.}$ The
use of such multi-tracer dust deposition flux estimate was deemed more robust to account for
variability of individual tracers in aerosols.
When using the average UCC metal composition, mean $F_{lithAv}$ of 1.8 mg m$^{-2}$ d$^{-1}$ calculated in this
study across the 2016-2021 time-series is consistent with earlier lithogenic deposition fluxes
reported in the literature. Dust peaks were consistently observed during the austral summer
(December to February), reaching fluxes up to 9.2 mg m$^{-2}$ d$^{-1}$ and low $F_{lithAv}$ fluxes down to 0.06
mg m$^{-2}$ d$^{-1}$ were estimated in the wintertime. Overall, individual year $F_{lithAv}$ fluxes also aligned
with the occurrence of known dust and bushfire events in the summertime as well as other global
meteorological events such as El Niño Southern Oscillation (ENSO).
$F_{LithAv}$ estimated using Al, Fe, Th and Ti content in the average UCC may present an advantage in
being more comparable with other studies worldwide. However, our $F_{LithAv}$ estimates (0.09 – 13.4
mg m$^{-2}$ d$^{-1}$) using Australian soil data as a crustal reference showed better agreement with mean
lithogenic fluxes reported by global modelling studies. Therefore, dust deposition estimates
calculated using local soil composition data are recommended for validating model outputs as they
likely better represent true deposition fluxes for our study region.
Dust emissions and deposition remain poorly quantified in global atmospheric models (Ito et al.,
2020). Therefore, our study reports precious field-based dust deposition flux estimates which are
essential to better constrain and validate modelling outputs, especially for Southern Hemisphere
dust sources (including Australia) which greatly vary in nature and composition. A wide range of
sampling methods should be used, including sediment core, sediment trap and aerosol sample
analysis, for which a multi-tracer approaches may be favoured when calculating lithogenic fluxes
compared to a single tracer approach. Samples covering a wide geographical area as well as
temporal (including time-series stations and winter period sampling) coverage are required to
better constrain seasonal and interannual variability. Meteorological parameters, isotope analysis
and modelling can also help better constrain the origin of lithogenic particles observed in field-
based studies.



## Author contributions

A.R.B. was responsible for project conceptualisation, funding acquisition, resources and supervision. M.M.G.P. was responsible for part of the sample collection, sample processing, data interpretation, processing and curation as well as for manuscript drafting. S.M was responsible for part of the sample collection and analysis, and for laboratory supervision. T.H was responsible for data curation. C.N was responsible for the analysis of soil samples. C.H was responsible for part of the sample collection, sample processing, data curation and the original draft writing. A.T. was responsible for instrumental analysis. P.dC. was responsible for part of the sample collection and data curation. M.S. was responsible for part of the sample collection and sample processing. All authors were responsible for data interpretation and validation and reviewing and editing the manuscript.

## Conflicts of interest

There are no conflicts to declare.

## Acknowledgements

A.R.B would like to thank the Australian Research Council (ARC) for part funding this work under grants FT130100037 and DP190103504. The Australian Antarctic Program Partnership (AAPP) is also acknowledged for support of laboratory costs as part of the Antarctic Science Collaboration Initiative (ASCI000002). Access to ICP-MS instrumentation was made possible through ARC LIEF funding (LE0989539). M.M.G.P was partly supported by ISblue project, Interdisciplinary graduate school for the blue planet (ANR-17-EURE-0015) and co-funded by a grant from the French government under the program "Investissements d'Avenir" embedded in France 2030. Soil samples were provided by The South Australia Drill Core Reference Library and the Geological Survey of South Australia, within the Department for Energy and Mining; many thanks to Anna Petts for assisting with legacy soil data selection and retrieval. The National Geochemical Survey of Australia, which provided the topsoil samples from Western Australia, South Australia, and Victoria, was a collaboration between Federal, States, and Northern Territory geological surveys led by Geoscience Australia (GA) and funded by the Australian Government's Onshore Energy Security Program (2006-2011). We thank GA for making those samples available for the present

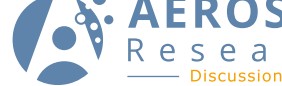

study.

## Acknowledgment to country

Before the white settlement of lutruwita/Tasmania, kunanyi/Mount Wellington was a prominent
feature in the lives of the Moomairremener people for thousands of years and continues to be. We
pay our respects to elders' past, present and emerging and are thankful to have been able to study
this region.

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
