# Peer review of "On the use of lithogenic tracer measurements in aerosols to constrain dust deposition fluxes to the ocean southeast of Australia."

_Aerosol Research, 2024_

## Author Comment (AC1)

**Dust deposition fluxes at the gateway to the Southern Ocean: investigating the use of lithogenic tracer measurements in aerosols collected in Tasmania, Australia**

https://doi.org/10.5194/ar-2024-21

**Response to reviewers**

Reviewer 1: Zongbo Shi

This is a solid paper. It proposed a multi-tracer method to estimate dust fluxes. It then estimated the dust flux at a strategic location in the Southern Ocean. The methodology is robust. The results are well presented and the conclusion is well justified.

I only have a few minor comments for consideration.

1. Title – be more concise. Words like "investigating" are a waste of space

Good suggestion, thank you. In addition, a second Reviewer suggested to refine the geographical position of the study location in the title. Amended title to: "On the use of lithogenic tracer measurements in aerosols to constrain dust deposition to the ocean southeast of Australia ".

2. Line 12-12: I suspected that you mean the flux estimated is between a peak dust deposition event and a low event. If so, the writing as it is does not represent this. Please clarify this and revise accordingly.

Changed to: "Lithogenic flux estimates showed annual dust deposition maxima during austral summer, following the Australian dust storm season, and annual minimum deposition flux over winter." lines 11-13 of the revised manuscript

3. Line 16-19: This is a bit wordy. The first part of the sentence appears to repeat the previous sentence. The main point appears to be something like: the data provided here will help to constrain model estimates of ….

Changed to: "The data provided here will help to constrain model estimates of southern hemisphere atmospheric deposition fluxes and their subsequent impact on global ocean biogeochemical cycles." lines 13-15 of the revised manuscript

4. Line 136: explain why 125 samples only for 6 years? E.g., give information on the sample duration and frequency.

The sentence at lines 136-141 has been removed and inserted above after the sentence at line 113-116, which explains sample duration/frequency. Additional information was also added at lines 118-120 of the revised manuscript regarding the number of samples included in this study. "Samples suspected for contamination or that were significantly wet at the time of recovery were discarded and sampling was suspended in the winter time of 2017, 2018 and 2019."

5. Line 196-197: how an aliquot be DRY sieved?

Changed to: 'Ten milligrams of each soil sample was dry sieved…" line 211 of the revised manuscript

6. I am sorry if I have missed but how the total mass of aerosols was estimated? This is important to mention as it determines the accuracy of the Fe/total aerosol mass ratio. It should be noted that there may well be sea salt and other natural/anthropogenic aerosols. This could reduce the total Fe content in the total aerosol. Similar applies to Al. This may partially explain the low mean Al/Fe contents in aerosols. It would be great if a mass closure (e.g., sulfate, nitrate, sea salt, OC, EC, dust etc.) is given if such data are available. This comment is also relevant for points raised in the paragraph starting line 286.

In the Methods section, line 168 onwards, the deposition flux calculation is detailed. As mentioned lines 186-189, we used the "Similar to other studies reported in the literature, a single-tracer dust (lithogenic) deposition flux estimate, $F_{Lith(X)}$, was calculated by dividing F(X) by the average abundance ($[X]_{UCC}$, wt%) of the element X in the UCC as reported in McLennan (2001); Al = 8.04%, Fe = 3.5%, Th = $1.07x10^{-3}$%, Ti = 0.41% following equation (2)". That way, the total aerosol mass was estimated using the relative abundance of each tracer in the upper UCC as per McLennan (2001).Unfortunately no additional measurements other than trace metals were available for all samples of the time series dataset. That way, no direct calculation on the total aerosol mass was possible in this study.

Also added lines 169-171 of the revised manuscript: "Additional measurements on the collected aerosols (e.g.; carbon and major ion analysis) were not available for this study, so intrinsic calculation of the total aerosol mass on each individual aerosol filter using these parameters was not possible."

7. Table S2 – total Fe content in soil appears to be very low. Yes, there may be spatial variabilities. But could there also be a possibility of the size dependence? The size cut here is about 63 um. And in reality, you are unlikely going to see many particles of that size at the sampling location due to long range transport (not to say that it is impossible). I suggest that the authors look at literature and see how other studies have estimated the total Fe content, both in terms of the size cut of the particles, and methodology. Secondly, can you show all other elements you measure for all soil samples. They are very useful reference data for future research.

Methodology aimed at representing particle grain size capable of being uplifted from the source region and transported as aerosol towards our aerosol sampling site at Mt Wellington. Our methodology matches other studies undertaken in Australia as Strzelec et al 2020 (see lines 211 - 212) as well as the National Geological Survey Australia size cut for soil fine fraction ( <75um, de Caritat et al., 2009).

A Table S3 was added to the supplementary documents, including measurements of other trace metals in the selected NGSA soil samples. This data will not be discussed in the main manuscript as it is out of the scope of our study. Mention to the new table S3 was added lines 243-245 of the revised manuscript "Metal concentrations in individual NGSA soil samples analysed in this study are reported in the supplementary Table S2 (lithogenic tracers) and in Table S3 (other analysed trace metals not discussed in this study)."

8. Line 266: this is an interesting point. Later sentences supported this argument. Are there representative back trajectories that you can show to support this point? It would be good to have the back trajectories from high and low dust flux seasons.

HYSPLIT air-mass trajectory frequency (AMBT) analysis was added to the supplementary documents Figure S1, in place of the initial Figure S1. Using the HYSPLIT AMBT model shows no significant seasonal difference in the observed wind influences at our sampling station. This may be due to the coarse resolution of the model which emphasises the prevailing westerly winds originating from the Southern Ocean. However, contrasting information is shown using mean wind data from Australian Bureau of Meteorology (BoM) for our sampling station (shown in a new supplementary Figure S2). Figure S2 seems to show increased winds originating from the North (from the Australian mainland) in the months of January through to March.

As both observations from HYSPLIT and from BoM diverge, further discussion was added lines 283-289 of the revised manuscript :

"While enhanced air-masses originating from the Australian mainland cannot be observed in the summertime using HYSPLIT model (supplementary Figure S1), the Australian Bureau of Meteorology reports increasing southwards blowing winds at our sampling station from January through to March (supplementary Figure S2). Such discrepancies emphasise the complex regional wind pattern influencing our sampling station and highlight the need to consider other parameters such as seasonal changes in environmental conditions at the source region when investigating aerosol entrainment and transport. "

In addition, literature supporting the south eastwards transport of dust from Australian mainland is referred to in the manuscript :

Mackie D.S. Biogeochemistry of iron in Australian dust: From eolian uplift to marine uptake. *Geochem. Geophys. Geosyst.* **2008**, *9*, Q03–Q08. doi.org/10.1029/2007GC001813

Baddock M., K. Parsons, C. Strong, J. Leys, G. Mctainsh, Drivers of Australian dust: a case study of frontal winds and dust dynamics in the lower lake Eyre basin. Earth Surf. Process. Landforms, 40 (2015), pp. 1982-1988, 10.1002/esp.3773

Che Y., B. Yu, and K. Bracco, Temporal and spatial variations in dust activity in Australia based on remote sensing and reanalysis data sets, EGUsphere [preprint], https://doi.org/10.5194/egusphere-2023-1710, 2023.

Tang W., J. Llort, J. Weis *et al*. Widespread phytoplankton blooms triggered by 2019–2020 Australian wildfires. *Nature* 597, 370–375 (2021). https://doi.org/10.1038/s41586-021-03805-8

9. Line 316 – spelling error

Changed 'ound' to 'found' in the revised manuscript

10. Figure 3 – please mention briefly what ratios are being used? UCC or Australian soil results?

This is already stated in the final sentence of the caption.

11. In Figure 1, would it be appropriate to consider add the locations by Strzelec et al. (and any other studies) where the dust flux was estimated?

In the revised manuscript, Figure 1 now displays the study location of previously reported dust deposition fluxes.

12. There are mentions of fire and related dust. This is an interesting point but I do wonder whether you can provide any supporting evidence, such as higher K+ concentrations. I presume you haven't analysed levoglocosan?

Levoglucosan or potassium analysis are beyond the scope of this study. Due to the lack of evidence of the proposed hypothesis, we decided to remove the suggested impact of fire emissions and only rely on NGSA database supporting higher Fe/Al ratios in Tasmanian soil. Amended sentence line 296-300 of the revised manuscript "Summertime Fe/Al ratios in kunanyi/Mt Wellington aerosols were slightly higher (Fe/Al =0.72) than the Australian soil measurements. This can be explained by increased contribution of local soil emission from Tasmania under drier weather conditions as the NGSA database shows higher Fe/Al ratio (on average 0.7, n=21 samples) in Tasmanian soil compared to other soil across Australia (Caritat and Cooper, 2011; Caritat, 2022)."

13. Paragraph starting 397: I wonder whether you can compare the estimated fluxes with more modelling studies. I think there are several global modelling studies of dust deposition fluxes. For example, Mahowald et al. 2005. https://agupubs.onlinelibrary.wiley.com/doi/10.1029/2004GB002402

The Mahowald et al. (2005) reference was not included in the manuscript as it was judge to be a contemporary literature to Jickells et al. (2005) and reporting similar dust fluxes (1.4- 2.7 mg/m$^{-2}$/d$^{-1}$). The dust flux reported in Mahowald et al. (2005) falling within the 1.4 – 5 mg/m$^{-2}$/d$^{-1}$ range now discussed in the revised manuscript line 370, the suggested literature was added to the reference list

Mahowald, N. M., Baker, A. R., Bergametti, G., Brooks, N., Duce, R. A., Jickells, T. D., et al. (2005). Atmospheric global dust cycle and iron inputs to the ocean. *Global Biogeochemical Cycles*, **19**(4), GB4025. https://doi.org/10.1029/2004GB002402.

14. Uncertainties: I agree that the multi-tracer method is more reasonable and better than a single tracer estimate. However there are still uncertainties and I suggest that you add a paragraph or section to discuss specifically about all the possible uncertainties, e.g., ratios, deposition velocities. It does not affect the conclusion of this paper but it will help readers to understand the manuscript better, and to use it more appropriately. If you can give an estimate of the uncertainty range, e.g., 2 times, that would be helpful. But I suspect it is not going to be an easy job. It may be worth mentioning that models still have large uncertainties so the uncertainties from observation-based flux estimates are still relatively small.

Uncertainty associated with the use of a set deposition velocity in our flux calculation was already mentioned in the manuscript Materials and Methods section although a quantitative uncertainty is now provided lines 180-184 of the revised manuscript "It should be mentioned that a factor of 3 uncertainty was previously attributed to the use of a set deposition velocity as it does not account for specific particle size in different aerosol samples or for specific atmospheric conditions such as humidity and wind speed at the collection time (Baker et al., 2016; Winton et al., 2016; Duce et al, 1991)."

Uncertainty associated with the use of the lithogenic tracer's abundance in the UCC (as a mean to calculate total aerosol mass) is also mentioned in the paragraph starting line 483 onwards of the revised manuscript. A paragraph summarizing uncertainty surrounding our dust flux calculation was added to the conclusion of the revised manuscript, lines 507-514. "Dust deposition fluxes calculated in this study hold some uncertainties of a factor 3 and a factor 2 due to the use of a set deposition velocity and the assumption of metal abundance as per the average UCC, respectively …"

15. Conclusions – this is rather long. Most of the points in the conclusions have already been mentioned in abstract. I wonder whether the final section as "Atmospheric implications" might be more useful to readers. Can you tell us a bit more about the implications of the results reported here? You mentioned nutrient inputs – are the inputs, in different seasons, likely to be important for ocean plankton and biological pump?

As an abstract is a condensed version of a complete paper, the authors feel it is reasonable to have some repeated conclusions in the Abstract and Conclusions sections. However, we have re-written the conclusion to make it shorter and more readable. Implications of the study are further developed in the final paragraph.

16. Can you also have a short paragraph, perhaps at the end of the "atmospheric implications" if you decide to have one, about what future research should be

done? You did mention somewhere in the text about the research needs – but they could be at one place of the manuscript.

Future research needs, including but not restricted to additional time-series data and winter samples, are now mentioned in the last paragraph of the conclusion section.

---

## Author Comment (AC2)

**Dust deposition fluxes at the gateway to the Southern Ocean: investigating the use of lithogenic tracer measurements in aerosols collected in Tasmania, Australia**

https://doi.org/10.5194/ar-2024-21

**Response to reviewers**

Reviewer 2: Anonymous

In this article, Hird et al. investigate the potential of using the concentration of four trace metals (Al, Fe, Th, and Ti), either independently or simultaneously, to estimate dust fluxes at the kunanyi/Mount Wellington time-series station, a site located in the wind corridor that transports Australian dust to the Southern Ocean. The authors present and analyze a data series collected over a period of 5 years, identifying an apparent seasonal behavior, and concluding that the variability in the calculated dust fluxes is reduced using one multi-elemental approach. Although it is a highly valuable dataset, the authors need to address key issues in their manuscript before it can be considered for publication.

**General Comments**

1. From my personal perspective, the title of the work needs to be adjusted according to what is presented. According to the title, dust deposition at the entrance to the Southern Ocean will be quantified, exploring the use of lithogenic tracers in aerosols collected in Tasmania, Australia. However, since the Southern Ocean has many entry points, the title is a bit too broad for the scope of the manuscript. Furthermore, compared to what is stated in the title, instead of justifying and demonstrating that tracers work for calculating dust fluxes, the manuscript focuses more on the calculation of enrichment factors of metals in particles, their comparison with continental soils, and the seasonality of the calculated deposition fluxes (see the first sentence of the conclusions). This first comment is very important, as I believe the authors need to better focus their work.

We agree that the Southern Ocean is broad and that our study only encompasses a small part of it. On that note, we suggest a new title for the manuscript as follows: "On the use of lithogenic tracer measurements in aerosols to constrain dust deposition to the ocean southeast of Australia ".

However, the scope of the study is, indeed, to propose the use of the multi tracer approach in aerosol studies. Indeed, the EF calculation is solely used to ensure the lithogenic nature of the tracers used (namely Al, Fe, Th and Ti). Similarly, analysis of the seasonal and inter-annual fluctuations of the estimated lithogenic flux at our sampling station is not interpreted in depth in this study. Rather, FLith are compared to major climatological seasons (e.g., dust season) and compared to specific events (large fire

emissions or annular modes) known to influence dust deposition to validate the robustness of our observations.

2. As they mention in the manuscript, they considered the proposal by Traill et al. (2022), who used these four tracers to calculate atmospheric flux in material collected in sediment traps at 1000 m depth. In the cited work, it is understood that the material consists of particles of various origins collected in a very complex environment, where the exposure time of the collectors and the collection area are clearly known. However, in the current work, why is it important to use this tracer approach if there is a way to accurately quantify the collected mass and it is certain that the collected material is aerosols? As I understand it, the greatest uncertainty in calculating a deposition flux arises when considering a constant deposition velocity, like the one you used (2 cm/s). However, by using the chemical composition of lithogenic metals as a tracer, you are not avoiding considering a constant deposition. So, what is the rationale for exploring the use of metal concentration in aerosols to estimate their mass and deposition? Isn't it simpler to quantify the mass of aerosols by weight difference rather than performing acid digestion and measuring metals by ICP-MS? I would like you to mention in your introduction the advantages this method has over others and, in your discussion, compare the flux estimation using other methods, including gravimetric analysis.

In our current work, we propose a method for studies looking into aerosol trace metal composition to estimate a more robust dust deposition flux than the commonly reported Al-based dust flux. Indeed, while the study of aerosol filter composition often includes the assessment of aerosol total loading by a gravimetric technique; additional processing steps involved in this later method (at least 24 h in a desiccator prior to and after sampling and handling of the filter using tweezers for weighing on a precision scale) is not compatible with trace metal analysis as it adds to the risk of contamination of the sample. This is especially true for aerosol filters collected in pristine environments such as the southern hemisphere ocean and coastal areas. Another issue encountered with particle collection in pristine environments is that the mass loading of aerosol may be too small to be accurately weighed on a precision scale, resulting in an associated error too large to consider gravimetric techniques. Finally, the total aerosol loading mass as provided by gravimetry does not allow distinguishing dust from anthropogenic sources, which can be done using specific tracers of lithogenic sources.

A sentence was included in the introduction of the revised manuscript line 80-84: "Different methods have also been used to estimate dust deposition fluxes from field samples. While broadly used in air pollution studies (Bindu et al., 2016), total aerosol loading measurements based on gravimetry does not enable the discrimination of atmospheric sources (e.g., dust vs anthropogenic). In addition, such a method is not compatible with atmospheric trace metal studies, where strict protocol requires minimal filter handling to prevent contamination. "

3. In the methodology section, you should expand the information on dust collection and the calculation of fluxes using a hivol device:

- 1) sampling time, explaining why that time was used and, in case of high variability between samples, discussing why and how it affects the results. Currently, something unclear is mentioned (L115: "with each sample representing a period ranging from a few days to 2 weeks");

Details on the time of aerosol sampling was already provided in the initial manuscript. Text was moved up to section 2.1 (lines 115 - 120 of the revised manuscript) for an easier read under the request of Reviewer 1.

- 2) Was the collected mass determined by gravimetry using the hivol device?;

The HiVol device does not allow an analysis of total aerosol mass by gravimetry itself. The choice for not choosing this method are developed in the response to Reviewer 2's question 2 above already (including trace metal clean procedures and the non selectiveness of source for gravimetric measurements).

- 3) Were the filters cut for digestion?;

Information was added on the sampled filter size (line 141) and on the subsampling method (lines 151-152) in the revised manuscript. This sub-sampling method was validated in an earlier study by Perron et al. (2020).

"One sub-sample of 47mm diameter was cut off each aerosol filter sheet collected at our sampling station using a sharp titanium punch cutter (Perron et al., 2020)."

- 4) How is the "total metal concentration (ng m-3)" calculated, as it is unclear?;

Because our leaching protocol is sequential, the total concentration of a metal X in the sample is obtained as "the sum of all three steps (of) the protocol". This information is displayed in the original manuscript.

To avoid any confusion lines 164-165 was added to the revised manuscript "Only total metal concentrations are discussed in the present study."

- 5) In the equations, the dimensional analysis is incorrect, making it difficult to understand how the fluxes were obtained. For example, in equation 1, considering the variables stated and the concentrations, you would not obtain a flux in mg/m2/s. The same issue occurs in equation 2, where the denominator is a concentration in wt%;

Equation 1 reported in the initial manuscript is correct and as follow :

flux = conc (ng/m3) x Vd (m/d)   (unit = ng/m2/d)

The resulting flux is expressed in ng/m2/d (not in mg/m2/s), which is the correct unit. Vd previously expressed in cm/s in the initial manuscript is now expressed in m/d to prevent any confusion, line 180 of the revised manuscript

Similarly, equation 2 initially presented shows the correct unit. The equation is displayed differently in the revised manuscript line 190 in order to prevent confusion :

$FLith(X) = (F(X) \times 100)/[X]UCC$

- 6) For clarity, the multi-elemental equation should be written;

It seems relatively simple to state that the FLithAv is the mean of all 4 individual FLith(X). However, following Reviewer 2's request, a third equation was added to the revised manuscript line 194. We, however, are not sure whether this addition makes the reading simpler or more complex.

- 7) I know it is mentioned later, but in the methodology section, it would be important to explain why the cold period is under-sampled, with only 2 out of the 6 sampled years represented. This last point is important for discussing whether seasonality truly exists or if it may be a result of under-sampling.

Answer to this question is included in the initial manuscript. Upon request from Reviewer 1, the text was moved up to lines 118-120 of the revised manuscript. "Samples suspected for contamination or that were significantly wet at the time of recovery were discarded and **sampling was suspended in the winter time of 2017, 2018 and 2019 for operational reasons.**" Operational reasons include power outage, road closure to get to the sampling station and water condensation in the sampler ; but we do not think such additional details are necessary for better understanding the study.

4. Another question that arises is, if the intention is to consider the concentration of lithogenic metals as proxies for the deposited dust mass, why weren't perchloric acid and higher temperatures (150 and 220°C) used during acid digestions? This could fully extract the metals from the particles and reveal their concentration. The absence of a total digestion protocol, like the one used by Traill et al. (2022), is concerning, as atmospheric fluxes could be underestimated. There is some evidence in the results that might point to this: 1) In Table S1, the recoveries presented for each metal are far from 100%; 2) if Table S2 is analyzed, the concentrations of Fe and Al in sieved top soils are low compared to the UCC, possibly indicating incomplete acid extraction; 3) something striking is what is presented in Table 2, as it would be expected that the best correlation would be between the elements with the highest abundance in the Earth's crust (Fe and Al).

We wish to thank the Anonymous Reviewer for this comment as it made us realize that some errors were displayed in the original Table S1, interfering with a good comprehension of the term "satisfactory recoveries" used in the manuscript. Indeed, amended Table S1 displays corrected recoveries for Al, Ti and Fe exceeding 90% while only Th recovery is at 76%. The later low recovery of Th is discussed in the initial and revised manuscripts. The protocol used in this study has previously been assessed and

validated for providing good recoveries for the studied lithogenic tracers (except Th). Data is displayed in Morton et al (2013) and Perron et al (2020).

Also, neither the initial or corrected recovery values does affect the discussion of lithogenic flux calculation using either the UCC value or Australian soil measurements. Indeed, observed differences between UCC and our Australian soil measurements mainly arise from differing Al and Fe concentrations for which recoveries exceed 90%.

Due to this mistake, full CRM data can be made available upon request for transparency purposes.

Literature:

M.M.G. Perron, M. Strzelec, M. Gault-Ringold, B.C. Proemse, P.W. Boyd, A.R. Bowie Assessment of leaching protocols to determine the solubility of trace metals in aerosols Talanta, 208 (2020), 10.1016/j.talanta.2019.120377

P. Morton et al. Methods for the sampling and analysis of marine aerosols: results from the 2008 GEOTRACES aerosol intercalibration experiment Limnol. Oceanog.: Methods 11 (2013), doi.10.4319/lom.2013.11.62

5. I would like to point out that the discussion of the seasonality of the calculated dust fluxes is not well supported. In fact, if you carefully observe Figures 2 and 3, half of the months in the time series were not sampled for the reasons mentioned by the authors. This creates a problem if the goal is to analyze any seasonal or interannual variation. What becomes evident is that the authors do not try to present meteorological evidence that helps compensate for the lack of data during the cold period. For example, it would be expected that a statement about seasonal variation in the calculated dust deposition fluxes would be accompanied by a detailed analysis of wind direction seasonality, relative humidity, atmospheric pressure, particle back trajectories, AOD, etc. The authors limit themselves to presenting a single graph of air-mass back-trajectory frequency that only considers 10 days before a single collection date, which of course is not representative of their entire study.

As stated above, the presented time-series does not aim at investigating neither the seasonal nor the inter-annual variabilities of dust flux at the study station. Rather, we present a method to dust deposition flux estimate. In doing so, we have to show consistency between our resulting fluxes and known seasonal or modal weather events which is done in the initial manuscript. Reviewer 2 highlights the lack of data in the winter time again ; this information is provided and acknowledged in the initial manuscript. Detailed information on atmospheric conditions at the sampling site were not available for this study as mentioned in the initial manuscript (see discussion on the deposition velocity constant). Monthly and seasonal air-mass back trajectories and local wind speed and direction data are now presented in the Supplementary Figures S1 and S2. This supplementary analysis emphasizes the complexity of the atmospheric

circulation at the sampling site. Further discussion was added lines 283-289 of the revised manuscript :

"While enhanced air-masses originating from the Australian mainland cannot be observed in the summertime using HYSPLIT model (supplementary Figure S1), the Australian Bureau of Meteorology reports increasing southwards blowing winds at our sampling station from January through to March (supplementary Figure S2). Such discrepancies emphasise the complex regional wind pattern influencing our sampling station and highlight the need to consider other parameters such as seasonal changes in environmental conditions at the source region when investigating aerosol entrainment and transport. "

In addition, literature supporting the south eastwards transport of dust from Australian mainland is referred to in the manuscript :

Mackie D.S. Biogeochemistry of iron in Australian dust: From eolian uplift to marine uptake. *Geochem. Geophys. Geosyst.* **2008**, *9*, Q03–Q08. doi.org/10.1029/2007GC001813

Baddock M., K. Parsons, C. Strong, J. Leys, G. Mctainsh, Drivers of Australian dust: a case study of frontal winds and dust dynamics in the lower lake Eyre basin. Earth Surf. Process. Landforms, 40 (2015), pp. 1982-1988, 10.1002/esp.3773

Che Y., B. Yu, and K. Bracco, Temporal and spatial variations in dust activity in Australia based on remote sensing and reanalysis data sets, EGUsphere [preprint], https://doi.org/10.5194/egusphere-2023-1710, 2023.

Tang W., J. Llort, J. Weis *et al.* Widespread phytoplankton blooms triggered by 2019–2020 Australian wildfires. *Nature* 597, 370–375 (2021). https://doi.org/10.1038/s41586-021-03805-8

Advanced small scale atmospheric modeling of the study region is out of the scope of this study.

**Other Comments**

**Abstract**

It is not clear what the actual contribution to the knowledge of aerosol fluxes to the Southern Ocean is. As I mentioned in the first general comment, the focus of the work needs to be improved.

This comment was attended to in the response to the major comment here-above. Manuscript title has been modified to better reflect the content of the manuscript

**Introduction**
L34-35, L45-47: These lines present similar information.

Original lines 45-47 were deleted in the revised manuscript

L100: See the first general comment.

Line 106: replaced 'the gateway' to 'one of the gateways'

**Methodology**

L140: When are the interannual trends discussed?

Differences in fluxes between years are discussed in sections 3.2 and 3.3 but are only used to validate our dust deposition flux calculation method. The aim of this paper is not to discuss details of the seasonal and inter-annual dust fluxes in the study region (this would require other atmospheric tracers measurements and modeling work far beyond the scope of this manuscript).

L143-144: The details of the chemical analysis should be provided in the manuscript. This includes a summary of the method detection limits, recovery percentages, measurement accuracy, and concentrations in the blanks.

The protocol used was assessed and validated previously and the reference to this study was cited in the original manuscript.
Literature:

M.M.G. Perron, M. Strzelec, M. Gault-Ringold, B.C. Proemse, P.W. Boyd, A.R. Bowie
Assessment of leaching protocols to determine the solubility of trace metals in aerosols
Talanta, 208 (2020), 10.1016/j.talanta.2019.120377

L146-149: Could the low recovery percentages be related to the leaching method? See general comments.

There was an error in the reported recoveries in the original manuscript which is now corrected in the revised manuscript as mentioned in the response to Reviewer 2 question 4.

L151: On what basis is a recovery considered satisfactory when it is greater than 80%?
This comment is addressed in the response just above and in the response to Reviewer 2 major comment #4 here-above.

L154-157: Considering the low recovery percentages, why did you continue to consider Th as a tracer for calculating atmospheric fluxes?

As discussed in the original manuscript, the variability of the resulting flux using all 4 tracers is very similar. The Th recovery presented in the original manuscript used a single non certified value for the ATD reference. The later ATD Thorium value is not part of the GEOTRACES consensus on ATD reference data. We decided to report this recovery value for Th in the supplementary table S1 for comparison purposes. However in the revised manuscript, we choose only to display Th recovery against a certified material. Using the GeoPT13 certified material, our Th recovery is 87%, this remains smaller than other lithogenic tracers due to the absence of HF in the analyzed leachate (as discussed in the original manuscript). We still defend that thorium can be used as a lithogenic tracer the same way as Al, Fe and Ti due to the very similar variability of its

dust flux derived estimate. It is acknowledged in our original manuscript that our protocol was not validated for Th analysis and that it could be improved in future study.

L161-164: Here, your Vd described in L169 should be mentioned. The same goes for lines L169-172, as they disconnect from L161.
The deposition velocity described lines 161-164 of the original manuscript refers to the one that is displayed line 169. This is a unique Vd of 0.2 cm s$^{-1}$. No other Vd value is used in our study.

For enhanced clarity, the value of the Vd used is also displayed in the revised manuscript lines 172-175 "Due to the lack of necessary meteorological data to estimate particle deposition velocities specific to our study site, a single coarse particle deposition velocity of 0.2 cm s$^{-1}$ was applied to trace metal-bearing dust deposition estimates based on the literature in similar study regions (Baker et al., 2017; Perron et al., 2020b; Winton et al., 2015)."

L167: The equation lacks data to fulfill dimensional analysis. Another question that arises is whether you also calculated particle flux using weight difference.

The incompatibility of the weight difference (gravimetric method) with trace element analysis in aerosols was discussed in response to several comments above. In addition, equation 2 was made clearer in line 82 of the revised manuscript for an easier read.

L177: If F(x) has units of mg/m2/d and [X]UCC is in %wt, what would the units of Flith(x) be? Could you clarify?
This comment was attended to in the major comment section. We confirm the correct unit was initially displayed for FLith (mg/m2/d) as the relative abundance of metals in the UCC (in percentage) is multiplied by 100 (%) so that only the F(X) flux unit (mg/m2/d) is to be accounted for in the resulting lithogenic flux unit (mg/m2/d). Equation 2 was modified for enhanced clarity line 190 of the revised manuscript.

L179: Could you clearly express exactly what you mean in an equation?
This comment was attended to in response to Reviewer 2 major comment. An equation 3 is now displayed line 194 of the revised manuscript although we do not think this equation brings much clarity into the manuscript.

L206-212: The use of the enrichment factor to identify sources is debatable. It is more suitable for understanding whether an element is enriched or depleted with respect to the upper continental crust.

This is exactly what enrichment factors are used for in this study : to confirm the lithogenic origin of our four "lithogenic" tracers. This is mentioned in lines 213-214 and lines 226-227 of the original manuscript. Later in the manuscript, elemental ratios (not EF) are used to assess resemblance with potential Australian dust emission source.

**Results and Conclusion**

See general comments.

L309-310, L315, L320: This does not indicate whether the correlation is significant. Statistical information is missing, and the coefficient needs to be properly named.

We agree on the misuse of the term "significant" in the original manuscript. Due to the minor aspect of the highlighted sentences in the manuscript, we removed the term significant from the highlighted sentences in the revised manuscript.

**Conclusions**

L149: When are the interannual trends discussed?

Differences in fluxes between years are discussed in sections 3.2 and 3.3 according to specific known events or mode influencing each year.

**Figures**

Significant improvements are needed in the figures. Among other things, the scales, shading, line thickness, font sizes, etc., in Figures 3 and 4 should be consistent. The same applies to the box and whisker plots.

All figures have now been standardized.

Figure 1 needs improvement—according to the results of the work, would seasonal changes in wind patterns be expected? According to the figure, are they dominant in a certain period of the year? The text is unclear, as it indicates several things and does not clarify what is being referred to. It is like a figure that the author must interpret. The acronyms in Figure 1 should be described in the figure caption. Avoid directing the reader to a table that is not presented in the main manuscript.

Figure 1 was adapted based on comments from Reviewer 1. Schematic dust pathway arrows have now been simplified for clarity. All information needed are in the caption of the new Figure 1 in the revised manuscript.

Figure 2: If elemental ratios are mentioned with exponents in Table S3, they should also be presented this way in the figure.

Elemental ratio numbers in Fig 2 have been modified to match format in Table S3

Delete Figure S1 due to its lack of contribution to the work, and consider including a real trajectory analysis in the main manuscript.

In the revised manuscript, Figure S1 was replaced by an in depth analysis of air mass back trajectory seasonal variability.

**Tables**

In Table 1, it would be better to represent concentrations in ppm and %, just as in McLennan (2001). This should also apply to Table S1.

In Table S1, all concentrations are now displayed in ppm and recoveries in %. Figure 1 already displays concentrations in ppm (1 ng mg$^{-1}$= 1 ppm). The displayed unit was changed to ppm in Table 1 caption for enhanced clarity.

In Table 2, are you sure that R2 is the correlation coefficient? Isn't it the coefficient of determination? I recommend using "R" . This also applies to the written text.

We can confirm that the value display is the R2 correlation coefficient between 2 named subsets of our database.

Table S1 should be moved to the main manuscript.

The method used to digest samples in our study was assessed and validated in a previous manuscript by Perron et al., 2020. It is not the aim of this study to assess the efficiency of the protocol applied again although recoveries are provided in the main manuscript. That way we think that Table S1 should remain in the supplementary materials

Literature:

M.M.G. Perron, M. Strzelec, M. Gault-Ringold, B.C. Proemse, P.W. Boyd, A.R. Bowie Assessment of leaching protocols to determine the solubility of trace metals in aerosols Talanta, 208 (2020), 10.1016/j.talanta.2019.120377